# Gut microbiome affects the metabolism of metronidazole in mice through regulation of hepatic cytochromes P450 expression

Nina Zemanová[1], Kateřina Lněničková[1], Markéta Vavrečková[1], Eva Anzenbacherová[1], Pavel Anzenbacher[2], Iveta Zapletalová[2], Petra Hermanová[3], Tomáš Hudcovic[3], Hana Kozáková[3], Lenka Jourová[1]*

1 Department of Medical Chemistry and Biochemistry, Faculty of Medicine and Dentistry, Palacký University, Olomouc, Czech Republic, 2 Department of Pharmacology, Faculty of Medicine and Dentistry, Palacký University, Olomouc, Czech Republic, 3 Laboratory of Gnotobiology, Institute of Microbiology of the Czech Academy of Sciences, Nový Hrádek, Czech Republic

* LenkaJourova@seznam.cz

**Data Availability Statement:** All relevant data are within the manuscript and its Supporting Information files.

## Abstract

Microbiome is now considered as a significant metabolic organ with an immense potential to influence overall human health. A number of diseases that are associated with pharmaco-therapy interventions was linked with altered gut microbiota. Moreover, it has been reported earlier that gut microbiome modulates the fate of more than 30 commonly used drugs and, vice versa, drugs have been shown to affect the composition of the gut microbiome. The molecular mechanisms of this mutual relationship, however, remain mostly elusive. Recent studies indicate an indirect effect of the gut microbiome through its metabolites on the expression of biotransformation enzymes in the liver. The aim of this study was to analyse the effect of gut microbiome on the fate of metronidazole in the mice through modulation of system of drug metabolizing enzymes, namely by alteration of the expression and activity of selected cytochromes P450 (CYPs). To assess the influence of gut microbiome, germ-free mice (GF) in comparison to control specific-pathogen-free (SPF) mice were used. First, it has been found that the absence of microbiota significantly affected plasma concentration of metronidazole, resulting in higher levels (by 30%) of the parent drug in murine plasma of GF mice. Further, the significant interaction between presence/absence of the gut micro-biome and effect of metronidazole application, which together influence mRNA expression of CAR, PPARα, Cyp2b10 and Cyp2c38 was determined. Administration of metronidazole itself influenced significantly mRNA expression of Cyp1a2, Cyp2b10, Cyp2c38 and Cyp2d22. Finally, GF mice have shown lower level of enzyme activity of CYP2A and CYP3A than their SPF counterparts. The results hence have shown that, beside direct bacterial metabolism, different expression and enzyme activity of hepatic CYPs in the presence/absence of gut microbiota may be responsible for the altered metronidazole metabolism.

**Funding:** This work was supported by the Czech Science Foundation (www.gacr.cz) [grant 19-08294S], which was awarded to TH and EA and the Internal Student Grant Agency of Palacký University (www.psup.cz) [grant IGA_LF_2020_022] awarded to NZ and MV. The funders had no role in study design, data collection and analysis, decision to publish, or preparation of the manuscript.

**Competing interests:** The authors have declared that no competing interests exist.

## Introduction

The cytochromes P450 (CYPs) are superfamily of heme-containing enzymes capable of metabolizing structurally diverse exogenous and endogenous substrates [1]. CYP enzymes are widely distributed in all kingdoms of life, from eukaryotic organisms such as animals, plants and fungi, to unicellular prokaryotic organisms–bacteria and archaea. CYPs are involved in oxidative biotransformation of most drugs and other various lipophilic xenobiotics [2]. Besides xenobiotic metabolism, they are involved in steroidogenesis and biosynthesis of many endogenous molecules important in regulation and function as e.g. vitamins (vitamin A and D), prostaglandins or thromboxanes [1].

The expression and function of biotransformation enzymes is multifactorially controlled including genetic and nongenetic factors such as gene polymorphisms, sex, age, disease, hormonal and diurnal influences [3]. These enzymes can be also affected by a complex microbial community in the gut, containing approximately 100 trillion cells also known as the gastrointestinal (gut) microbiota.

The gut microbiota is capable to affect the biotransformation xenobiotics directly or indirectly [4–6]. Direct influence includes a wide range of metabolic reactions. Another way in which gut microbiota can influence the metabolism, disposition and potentially toxicity of xenobiotics are indirect mechanisms. The indirect effects may include modulation of properties of host enzymes of metabolism and/or transporters and direct competition between host and microbial metabolism [4, 6]. To date, numerous studies have focused on the direct microbial modifications of drugs. For instance, previous research from this laboratory showed that nabumetone (widely used non-steroidal anti-inflammatory prodrug) is metabolized by bacteria to its non-active metabolites [7].

In the present study, the focus was on an antibacterial and antiprotozoal drug metronidazole (1-[2-hydroxyethyl]-2-methyl-5-nitroimidazole) (Fig 1). Metronidazole is one of the most prescribed medicines being one of the most used antibiotics in pregnancy [8]. It is highly effective against anaerobic infections (bone, joint, central nervous system and respiratory tract infections) [9, 10].

Antibacterial effect of metronidazole is based on its transformation to active metabolite inside the bacterial cell, in other words, it is a prodrug entering the bacterial cell unaffected. Subsequently, it is reduced to a nitroso free radical. Although this active cytotoxic metabolite may interact with many molecules, e.g. intracellular proteins, it is mainly the DNA that is

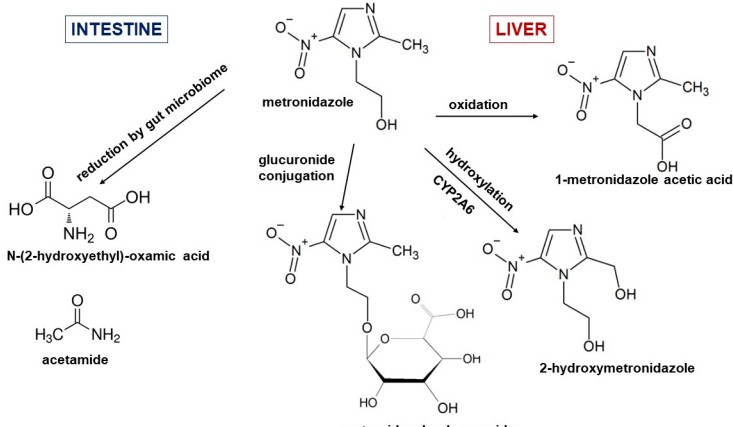

**Fig 1. Metabolism of metronidazole in humans.** In the human liver, metronidazole is metabolized by side-chain hydroxylation, oxidation or glucuronidation to form more polar metabolites. Metronidazole can also undergo modification by gut microbiota producing reduced metabolites such as N-(2-hydroxyethyl)-oxamic acid and acetamide.

affected by strand breakage, inhibited repair and disrupted transcription, which finally leads to death of the bacterial cell [10, 11].

In the mammalian organism, metronidazole, as a vast majority of xenobiotics, undergoes the biotransformation in the liver. It is mostly hydroxylation of the 2-methyl group (2-hydroxy-metronidazole), oxidation of the 1-ethyl group (1-metronidazole acetic acid) and glucuronide conjugation of hydroxylated metabolite (metronidazole glucuronide) (Fig 1). Along with its metabolites, metronidazole is eliminated in urine [12]. Even though metronidazole is for decades the drug of choice targeted against anaerobic bacteria and protozoa, specific processes related to its metabolization have not been fully characterized. However, it has been found, that CYP2A6 is the predominant catalytic enzyme forming 2-hydroxymetronidazole in humans [12].

As it has been mentioned earlier, gut microbiota is capable to affect the drug biotransformation. Metronidazole can undergo reductive modification by rat cecal contents or Clostridium perfringens, when enzymes of anaerobic bacteria disrupt the imidazole ring and produce reduced metabolites such as N-(2-hydroxyethyl)-oxamic acid and acetamide [13]. Moreover, it has been shown that reduced products of metronidazole can be detected in urine of conventional rats and not in germ-free rats [14], confirming that gut microbiota was required to modify metronidazole and, vice versa, that metronidazole can also alter the gut microbiota [15].

The aim of this study was to analyse the effect of gut microbiome on the fate of metronidazole in the mice through modulation the expression and activity of selected CYPs, in murine liver. To assess the influence of gut microbiome, unique germ-free (GF) mice in comparison to control, specific-pathogen-free (SPF) mice were used. The focus was on the CYP enzymes belonging to the 1, 2 and 3 CYP families, responsible for metabolism of the majority of drugs [1, 3]. To uncover the possible molecular changes leading to altered metronidazole metabolism in GF mice, the time course of the expression and enzyme activities was determined after metronidazole administration in comparison with the control SPF mice. Furthermore, the levels of the transcription factors involved in CYP expression (such as AhR, CAR, PPARα or PXR) [3] were analyzed in the both groups of mice.

## Materials and methods

### Chemicals

Prototypic substrates and metabolites used for the CYP enzyme assays, specifically diclofenac, bufuralol, coumarin, diazepam, 7-hydroxycoumarin and resorufin were supplied by Sigma-Aldrich (Saint-Louis, MO, USA). Fisher Scientific (Hampton, NH, USA) provided 4′-hydroxydiclofenac and 7-ethoxyresorufin. Desmethyldiazepam was supplied by Lipomed (Arlesheim, Switzerland). Coenzyme $NADP^+$, which was part of the incubation mixture, was purchased from Merck (Darmstadt, Germany).

All other chemicals (including metronidazole, 2-hydroxymetronidazole and paracetamol) of the analytical grade were obtained from Sigma-Aldrich.

### Animals

The experimental approach used in this study is a version of previous work [16]. The experiment consisted of control groups (SPF and GF mice without metronidazole applied) and groups of SPF and GF mice followed after the application of metronidazole (n = 4).

Two-month-old germ-free (GF) were used together with specific pathogen-free (SPF) males of inbred BALB/c mice of the same age. GF mice were born and housed under sterile conditions in Trexler-type plastic isolators; they were fed with sterile pellet irradiated 50 kGy diet Altromin 1410 (Altromin, Lage, Germany). The GF mice were monitored every week for the presence of contamination by standard microbiological methodologies. SPF mice were fed

with the same sterile diet as their GF counterparts. All animals were kept in a room with a 12 h light-dark cycle at 22˚C.

Metronidazole was administered as one intragastric dose (5 mg/kg) to both groups of SPF and GF mice. The anesthetized mice were decapitated, and livers were then harvested at different times—at 2nd, 6th and 24th hour after administration, weighed and kept frozen until the next procedure. Liver tissue samples for determination of mRNA expression were stored in RNA later® (Qiagen, Hilden, Germany).

Blood samples for determination of metronidazole in plasma were collected from the carotid artery into EDTA-treated tubes. The blood samples were centrifuged at 3200 × g for 10 min and plasma was separated and stored at -80˚C until use. The experiments were approved by the Committee for the Protection and Use of Experimental Animals of the Institute of Microbiology, Academy of Sciences of the Czech Republic (approval ID: 18/2019). The methods were carried out in strict accordance with the approved guidelines.

## Determination of metronidazole and 2-hydroxymetronidazole in murine plasma

Metronidazole and 2-hydroxymetronidazole were determined according to the method described earlier [16]. To each sample, internal standard (paracetamol) was added. Subsequently, 100% methanol was used to denature plasma proteins. Samples were then centrifuged, and supernatant was transferred to 1.5 mL Eppendorf tube and evaporated under nitrogen flow at 40˚C. The samples were dissolved in mobile phase and applied into a Li-Chrospher RP-18 column (Merck). For HPLC analysis Shimadzu LC-20 HPLC system (Shimadzu) with UV/fluorescence detection was used.

## Gene expression and quantitative real-time PCR

Total RNA was isolated, following the manufacturer's protocol, from murine liver tissues using RNeasy Mini Kit (Qiagen). Concentration of total RNA in each sample was quantified spectrophotometrically using the NanoPhotometer® N60 (Implen, Munich, Germany). RNA was then transcribed to single-stranded cDNA using Transcriptor High Fidelity cDNA Synthesis Kit (Roche, Basel, Switzerland).

The mRNA expression of transcription factors and selected CYP enzymes was quantified by real-time qPCR using commercial TaqMan Gene Expression Assays (Thermo Fisher Scientific, Waltham, MA, USA). Miniaturized qPCR in a 1536-well format plates were pipetted using Echo Liquid Handler (Labcyte, Dublin, Ireland) and performed on LightCycler 1536 Instrument (Roche). Calculations were based on the 2(-Delta Delta C(T)) method [17]. The values of target genes were normalized to the mRNA expression of housekeeping gene—hypoxanthine guanine phosphoribosyl transferase (Hprt).

## Liver microsomal fractions

Microsomal preparation was performed according to the established protocol [18] and fractions were stored at -80˚C. Concentrations of **CYP enzymes** were determined spectrophotometrically according to established method [19].

## Western blotting

Western blotting experiments were performed from pooled microsomal liver fractions. Protein concentration in each fraction was determined by the bicinconinic acid method using BCA protein assay kit (Sigma-Aldrich) according to the manufacturer's instructions.

Proteins of microsomal fractions (30 μg of protein was loaded per well) were separated using 4–15% Mini-PROTEAN TGX Precast Protein Gels (Bio-Rad, Hercules, CA, USA) and subsequently transferred onto polyvinylidene fluoride membranes (0.45 μm) using Trans-Blot Turbo Transfer System (Bio-Rad). Selected proteins were immunodetected with primary and secondary antibodies, according to method described in [20]. For the immunodetection of CYPs, following primary antibodies were used (cross-reacting with corresponding murine forms): CYP2C9 (Origene kat#CF503776, Rockville, MD, USA); CYP1A2 and CYP3A (Santa Cruz Biotechnology kat#sc-53614 and kat#sc-271033, Santa Cruz, CA, USA); CYP2B10 and CYP2D1 (Abcam kat# ab9916 and ab22590, Cambridge, United Kingdom). For detection of the immunocomplex, horseradish peroxidase-conjugated secondary antibodies and chemilu-minescence kit (Santa Cruz Biotechnology) were used. Image Studio™ Lite 5.0 (LI-COR Biosci-ences, Lincoln, NE, USA) software was used for the image analysis. Protein of glyceraldehyde 3-phosphate dehydrogenase (GAPDH) (Santa Cruz Biotechnology #sc-365062) served as the loading control.

## Cytochrome P450 enzyme activity assays

Enzyme activities were measured in the hepatic microsomal fractions by methods used in our laboratory according to the established protocols [19]. Selected enzyme activity assays were measured: CYP1A1/2 –ethoxyresorufin O-deethylation, CYP2A –coumarin 7-hydroxylation, CYP2B –pentoxyresorufin O-dealkylation, CYP2C –diazepam 3-hydroxylation, CYP2C – diclofenac 4′-hydroxylation, CYP2D –bufuralol 1′-hydroxylation and CYP3A –midazolam 1′-hydroxylation. Substrates of orthologous human forms were used. Incubation mixture for each individual enzyme activity contained potassium phosphate buffer (pH 7.4), NADPH-gen-erating system (NADP$^+$, isocitrate, isocitrate dehydrogenase and MgCl$_2$), liver microsomes and the respective substrate. For determination of metabolites, Shimadzu LC-20 HPLC system (Shimadzu, Kyoto, Japan) with UV or fluorescence detection was used. The analyses were per-formed with a LiChrospher RP-18 column (5 μm) 250×4 mm (Merck) The CYP3A activity assay (midazolam 1′-hydroxylation) was performed with a Chromolith® High Resolution RP-18 endcapped column (1.15 μm, 150 Å) 100×4.6 mm (Merck). The HPLC conditions–elu-tion and detection are given in Table 1.

## Statistical analysis

The statistical significance of gene expression was determined by two-way ANOVA using soft-ware IBM SPSS Statistics for Windows, Version 23.0 (Armonk, NY: IBM Corp.). Differences were regarded as statistically significant when the p-value was lower than 0.05. Software GraphPad Prism 8 (GraphPad Software Inc., California, USA) was used to create the graphs.

**Table 1. HPLC conditions for the measurement of enzyme activity assays.**

| CYP | Elution | Detection | |
|---|---|---|---|
| | | UV (nm) | Fluorescence Ex/Em (nm) |
| 1A1/2 | Isocratic | | 535/585 |
| 2A | Isocratic | | 338/458 |
| 2B | Isocratic | | 535/585 |
| 2C (diazepam) | Isocratic | 229 | |
| 2C (diclofenac) | Binary gradient | 280 | |
| 2D | Binary gradient | | 252/302 |
| 3A | Isocratic | 240 | |

Due to the scarcity of material, the statistical significance of activity assays could not be determined.

## Results

### Determination of metronidazole and 2-hydroxymetronidazole in murine plasma

In the preliminary work from this laboratory [16], the focus was on determination of the plasma concentrations of metronidazole and its primary metabolite 2-hydroxymetronidazole in SPF mice only. In the experiment described here, the difference in plasma concentrations of metronidazole and 2-hydroxymetronidazole between SPF and GF mice are demonstrated (Fig 2).

The time needed to reach the maximum level of metronidazole ($t_{max}$) was the same for both groups of mice (2 hours). In the second hour, concentration of metronidazole has been shown to reach the peak plasma concentration ($c_{max}$), with rapid decrease at 6th hour after drug administration (Fig 2A). In GF mice, plasma concentration of metronidazole was higher compared to SPF mice in both times (2nd and 6th hour). Maximal plasma peak concentration of metronidazole in GF mice was 148.13 ± 41.27 μmol/L and in SPF mice, it was 99.08 ± 35.74 μmol/L.

In SPF and GF mice, the time course of plasma concentrations of the metabolite, 2-hydroxymetronidazole, was almost identical (Fig 2B). The maximum level of 2-hydroxymetronidazole ($t_{max}$) was hence the same for SPF and GF mice (2nd hour) with plasma peak concentration ($c_{max}$) of 2-hydroxymetronidazole in GF mice 14.41 ± 3.78 μmol/L and 15.90 ± 5.18 μmol/L in SPF mice. The individual plasma concentrations of metronidazole and 2-hydroxymetronidazole are in S1 Data.

When the difference between levels of determined compounds was expressed as metabolic ratio (the ratio levels of unchanged drug to its metabolite), the differences between SPF and GF mice could be seen even better. The respective values were 6.23 (SPF mice) and 10.28 (GF mice) at the 2nd hour, and 3.07 (SPF mice) and 3.99 (GF mice) at the 6th hour after drug administration, respectively.

### mRNA expression of transcription factors

In all the qPCR experiments, the focus has been on differences in the mRNA expression and metronidazole-induced changes in the presence or absence of gut microbiota in liver samples. The data for SPF and GF mice were compared (four groups of mice– 0 = controls and 2nd, 6th and 24th hour after application of metronidazole). The amount of mRNA was expressed as relative expression.

Firstly, the mRNA expression of transcription factors involved in the regulation of various CYPs [3] were determined (aryl hydrocarbon receptor (AhR), constitutive androstane receptor (CAR), pregnane X receptor (PXR) and peroxisome proliferator-activated receptor alpha (PPARα)). Additionally, nuclear factor erythroid 2-related factor 2 (Nrf2), which is the key transcription factor in the regulation of protection against oxidative damage and has also an important role in regulation of phase I and phase II metabolizing enzymes, was analysed [21]. Interestingly, there is significant interaction between presence/absence of the gut microbiome and effect of metronidazole application, which together influence the mRNA expression of some transcription factors (Fig 3).

In the case of PPARα, the significant difference between basal mRNA expression of SPF and GF mice was found, when GF mice have shown approximately three times higher level of mRNA expression (Fig 3D). Moreover, PPARα exhibited a different time course of mRNA

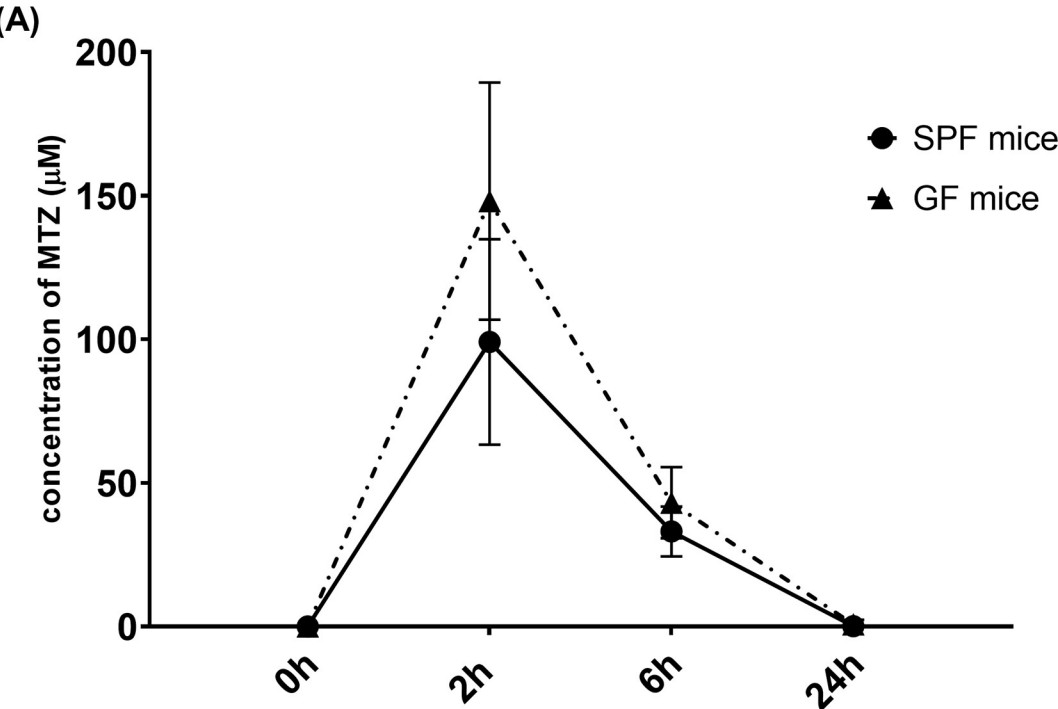

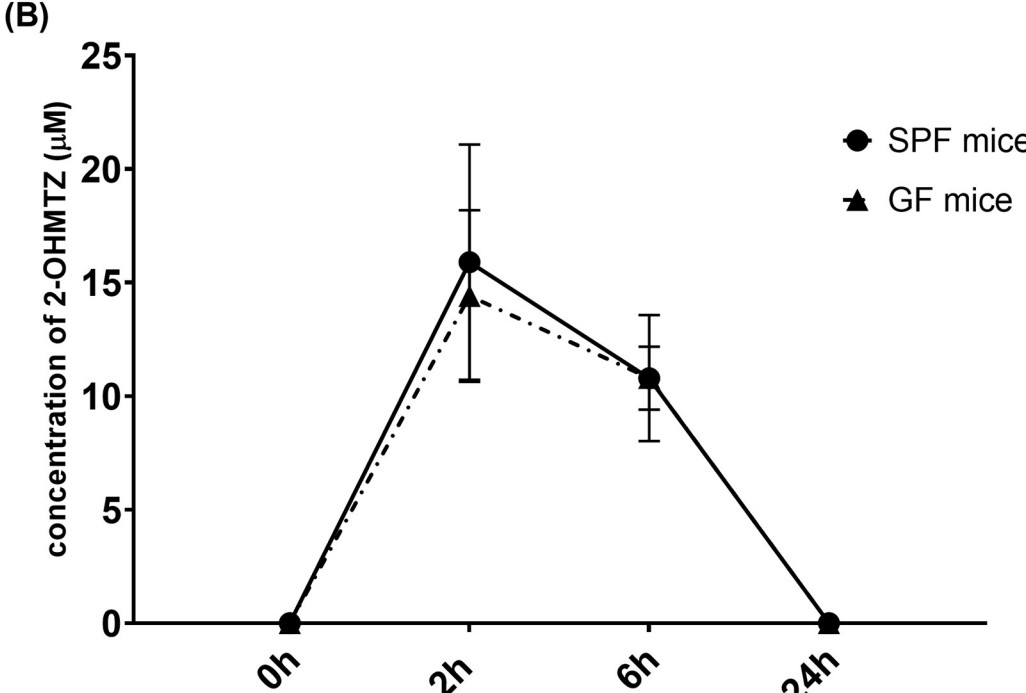

**Fig 2. Plasma concentrations of metronidazole and 2-hydroxymetronidazole in murine plasma.** Plasma levels of metronidazole (A) and its primary metabolite 2-hydroxymetronidazole (B) were determined at 2nd, 6th and 24th hour after administration to mice (the 0 hour taken as a control).

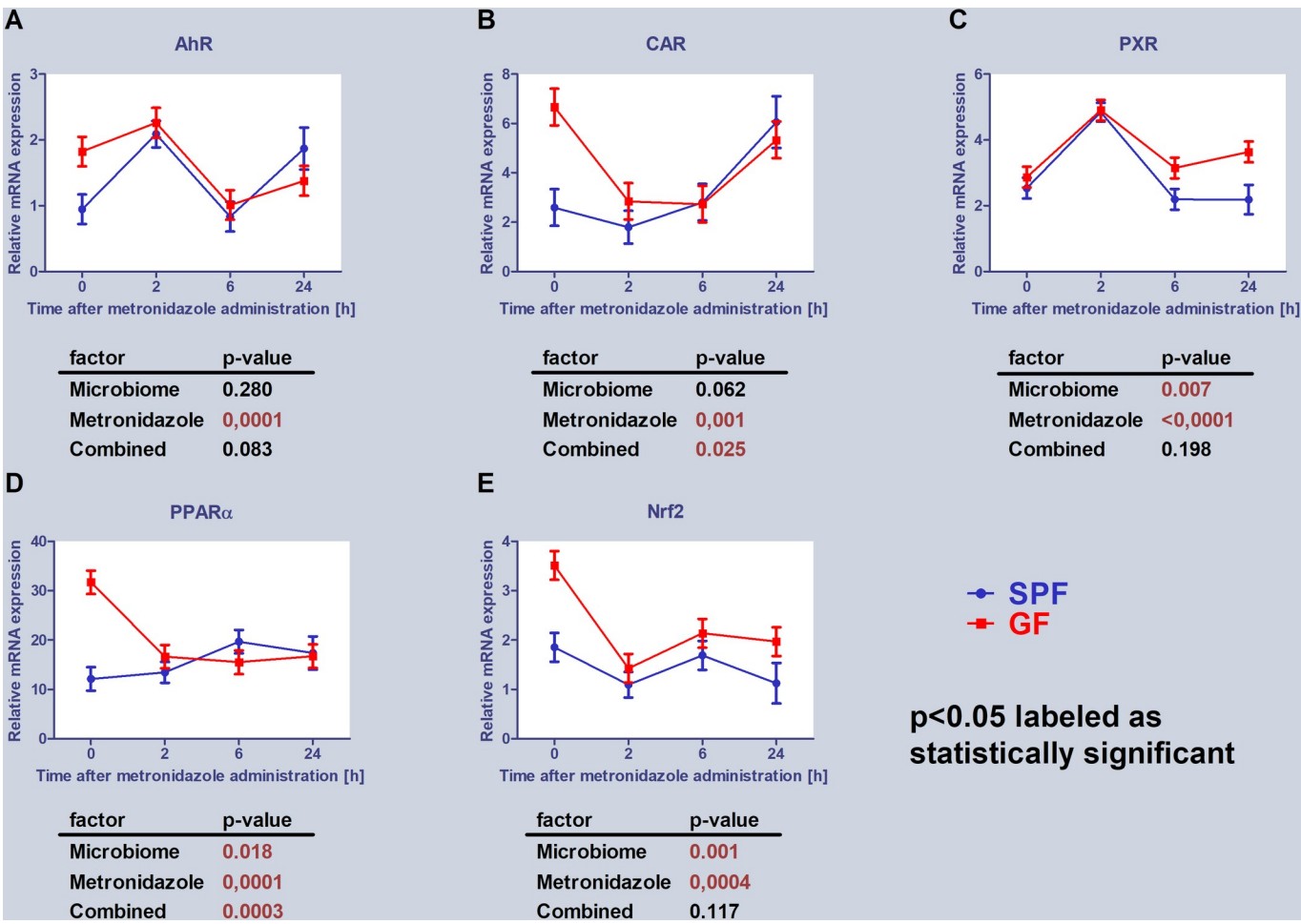

**Fig 3. Expression of transcription factors.** Comparison of estimated marginal mean values of mRNA expression of selected transcriptional factors in relation with microbiome presence and metronidazole administration in mice liver. The data represent the mean ±SEM from 4 individual animals. The statistical significance was determined using two-way ANOVA and p-values for the effect of microbiome (i.e., difference between SPF and GF groups), metronidazole administration and combined effect of the both are shown in the tables below the respective graphs. Significantly different from control p < 0.05.

expression after metronidazole application in the both groups. Finally, the significant interaction between presence/absence of the gut microbiome and effect of metronidazole application was determined here (Fig 3D).

The mRNA expression of CAR has shown significant interaction between presence/absence of the gut microbiome and effect of metronidazole application as well. Metronidazole administration itself also significantly affected mRNA expression of CAR (Fig 3B). Although, the basal mRNA expression of GF mice was higher than in SPF mice, taking together all groups, the difference between SPF and GF mice was not prominent.

Administration of metronidazole significantly influenced mRNA expression of other selected transcription factors–AhR, PXR and Nrf2 (Fig 3A, 3C, 3E). In the case of PXR and Nrf2, significant difference between GF and SPF mice was determined.

## mRNA expression of CYPs

The next qPCR experiment was focused on the effect of the gut microbiome (along with metronidazole administration) on the expression of murine CYP genes from 1, 2 and 3 families–

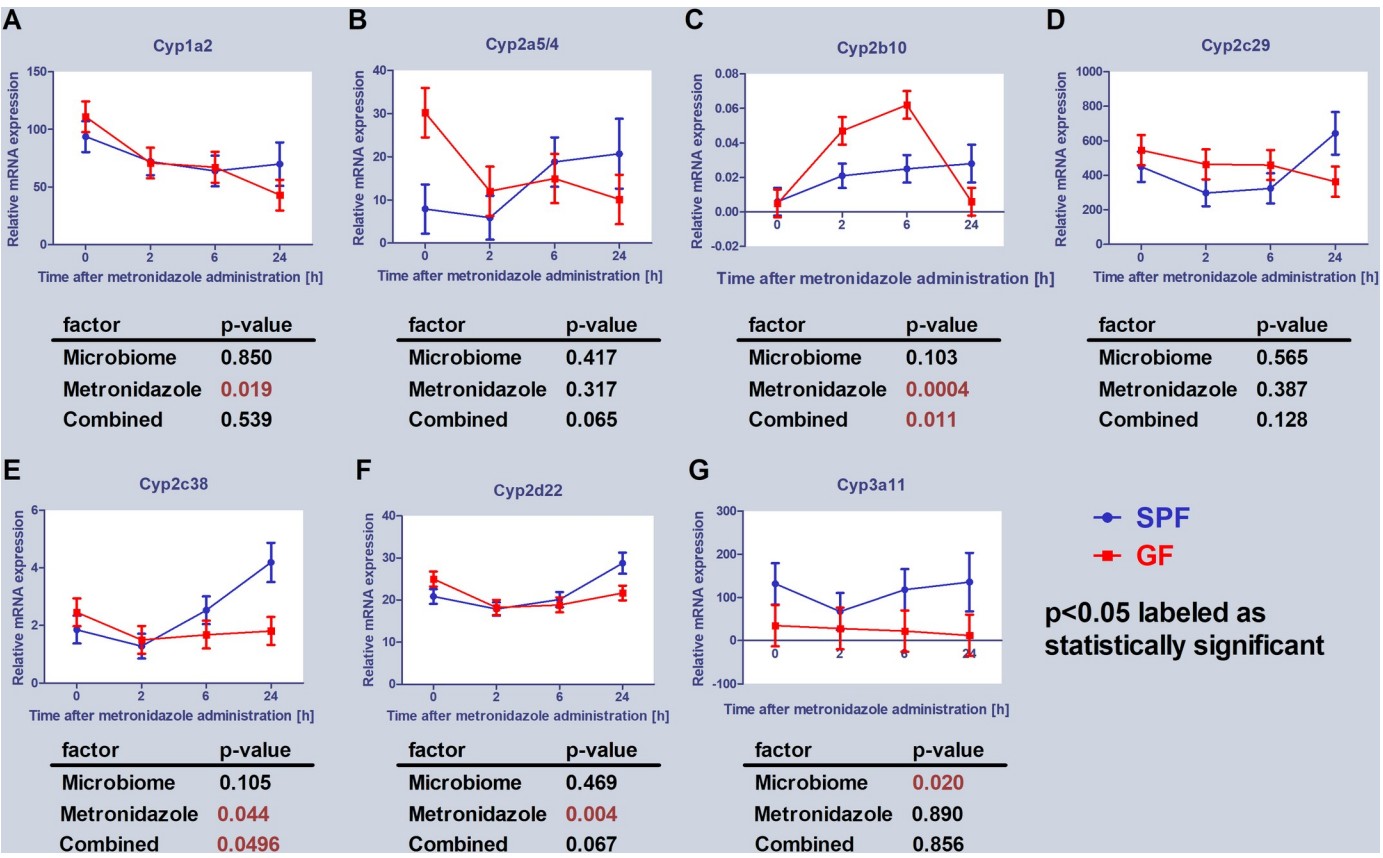

**Fig 4. Expression of CYP enzymes.** Comparison of estimated marginal mean values of mRNA expression of selected CYPs in relation with microbiome presence and metronidazole administration in mice liver. The data represent the mean ±SEM from 4 individual animals. The statistical significance was determined using two-way ANOVA and p-values for the effect of microbiome (i.e., difference between SPF and GF groups), metronidazole administration and combined effect of the both are shown in the tables below the respective graphs. Significantly different from control $p < 0.05$.

1a2, 2a5/4, 2b10, 2c29, 2c38, 2d22 and 3a11 (Fig 4), again, at various times after administration of the drug. The amount of mRNA was expressed as relative expression.

Significant combined effect of microbiome and metronidazole application was found in Cyp2b10 and Cyp2c38 mRNA expression (Fig 4C, 4E). Administration of metronidazole itself influenced significantly mRNA expression of Cyp1a2, Cyp2b10, Cyp2c38 and Cyp2d22. Only in the case of Cyp3a11 the significant difference between SPF and GF mice (the effect of microbiome) was observed (Fig 4G). Although, the effect of two factors (or their combination) was found insignificant in Cyp2a5/4 mRNA expression, the significant difference between basal mRNA expression of SPF and GF mice was found, when GF mice have shown more than three times higher level of mRNA expression (Fig 3B), which was significantly decreased by metronidazole application.

## Protein expression of CYPs

The effect of metronidazole on protein expression of five CYP forms (CYP1A2, 2B10, 2C9, 2D and 3A) was estimated by Western blotting in GF and SPF mice (Table 2, Fig 5).

Interestingly, in the case of CYP2B10, an increase of the expression of protein was found (Fig 5) in SPF mice (which was observed also for the mRNA). The western blotting was not able to distinguish between CYP2C forms, however, the protein expression of CYP2C9 (Fig 5) possibly followed an increase of Cyp2c38 mRNA in SPF mice (Fig 4A).

**Table 2. Relative protein expression (western blotting) of selected CYP forms.** Protein expressions at t = 0 h taken as control (equal to 1.00).

| | SPF mice | | | GF mice | | |
|---|---|---|---|---|---|---|
| | Time after application of metronidazole | | | | | |
| | 2nd h | 6th h | 24th h | 2nd h | 6th h | 24th h |
| CYP1A2 | 0.83 | 0.78 | 0.75 | 1.05 | 1.18 | 1.01 |
| CYP2B10 | 0.92 | 1.02 | 1.84 | 1.25 | 1.24 | 1.01 |
| CYP2C9 | 1.03 | 1.27 | 1.34 | 0.98 | 0.89 | 0.83 |
| CYP2D | 0.92 | 0.77 | 0.94 | 0.92 | 0.82 | 0.99 |
| CYP3A | 0.90 | 0.95 | 0.95 | 1.08 | 0.88 | 0.81 |

Samples obtained from the GF mice also indicated an increase of the expression of the CYP2B10 protein (Fig 5), reflecting an increase in the expression of the corresponding mRNA (Fig 4B); again, decrease of the protein expression after 24 hours after metronidazole administration was observed (as it was also the case with the corresponding mRNA of Cyp2b10 in GF animals). Protein expression of CYP1A2, CYP2D1 and CYP3A4 did not change significantly in the both groups of mice by metronidazole administration.

## Effect of metronidazole on CYP enzyme activities

The enzyme activities characteristic of CYP enzymes were determined in hepatic microsomal samples with substrates used in the corresponding assays [1, 19]. The data for SPF and GF mice were compared (0 = control, and time 2h, 6h and 24h after application of metronidazole).

For CYP2C, two substrates (diclofenac and diazepam) were used to cover the activities of characteristic for the most of the CYP2C forms involved. For the CYP2C38 activity, since it is similar to human CYP2C9 form [22], diclofenac was used as a substrate. The activity of CYP2C29 was determined by using diazepam as a substrate (murine CYP2C29 is homologous with human CYP2C19) [23].

Comparison of the data showed that in the case of the SPF animals, a trend of increasing enzyme activities of CYP2B and CYP2C (diazepam as substrate) in liver microsomes obtained from these (SPF) experimental models. Other CYP activities, prototypical of CYP1A, CYP2A, CYP2D and CYP3A did not exhibit significant changes (Fig 6). In the samples from the GF animals, there were no prominent or significant changes of CYP enzyme activities recorded by metronidazole application (Fig 6). GF mice have shown lower level of enzyme activity of CYP2A and CYP3A than their SPF counterparts (Fig 6B, 6G).

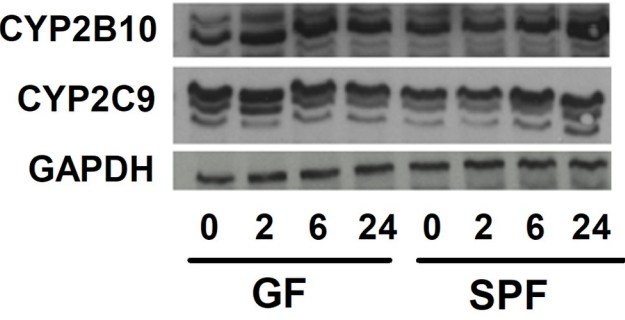

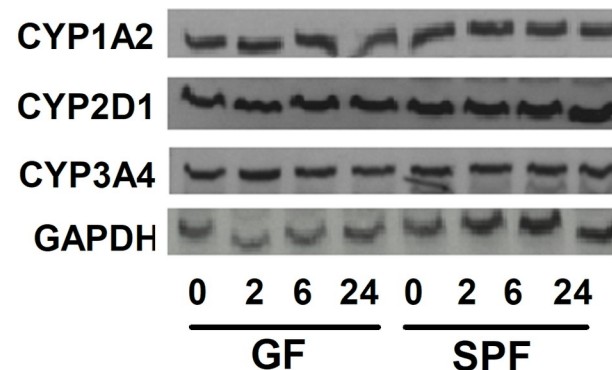

**Fig 5. Protein expression of CYP enzymes.** Protein expression was measured in pooled hepatic microsomal samples of four SPF and four GF mice (0 = control, and time 2h, 6h and 24h after application of metronidazole). Protein expression of CYP enzymes was normalized to the expression of GAPDH.

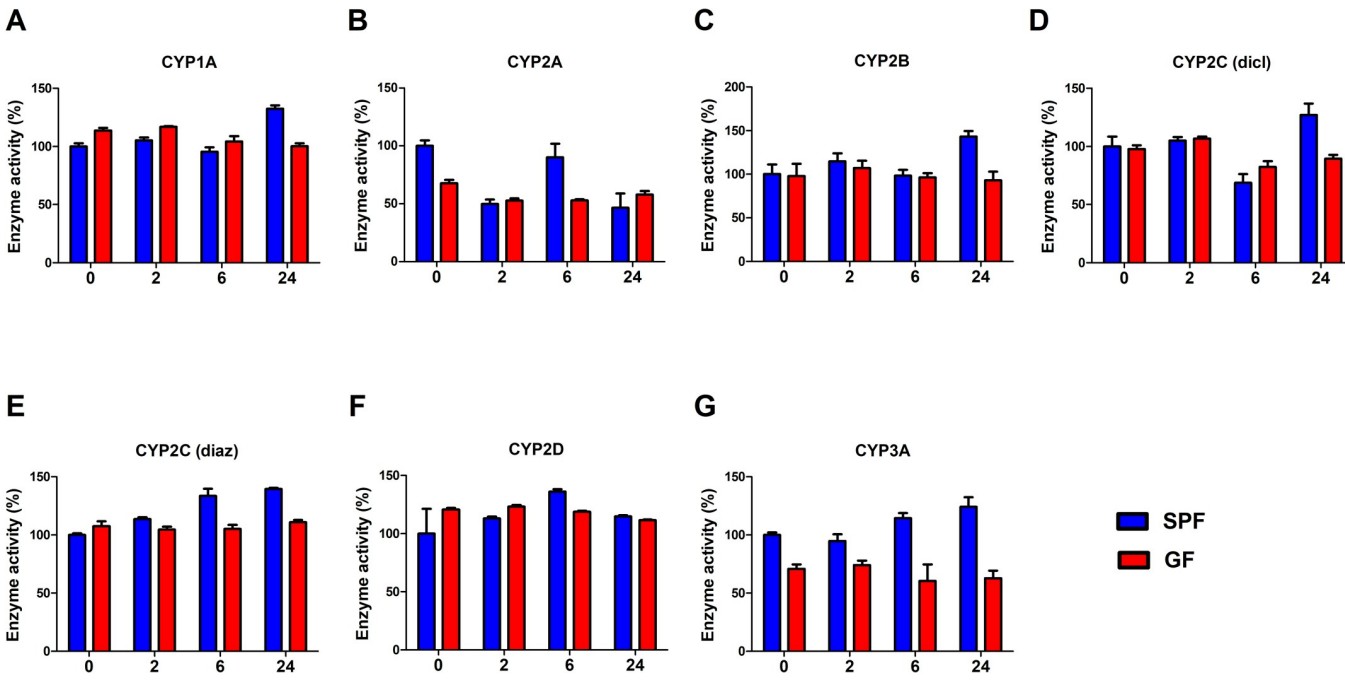

**Fig 6. Enzyme activity of CYP.** CYP enzyme activity was measured in pooled hepatic microsomal samples of SPF and GF mice (control group and time 2h, 6h and 24h after application of metronidazole). Data represent the mean ± SD from pentuiplicates measured in pooled microsomal samples of 4 SPF groups of mice and 4 GF groups of mice. Group of SPF mice before metronidazole application (0) was used as a control. (dicl–diclofenac, diaz–diazepam).

## Discussion

The ability of the gut microbiome to metabolize drugs was recognized over 50 years ago, in the report of Scheline on possible contribution of rat cecal microflora to metabolism of sulfonated azo dyes and phenolic acids [24] and many studies have provided recently sufficient evidence that this bacterial involvement in the biotransformation of clinically used drugs may alter their bioavailability, efficacy and toxicity [7, 25, 26]. Although, the gut microbiome metabolic activity may significantly contribute to the interindividual differences of response to pharmacotherapy, it belongs among the least explored factors contributing to this variability.

In this study, it was found that the absence of microbiota significantly affected plasma concentration of metronidazole, resulting in higher levels of the parent drug in murine plasma of GF mice. Plasma peak concentration of metronidazole was more than 30% higher in GF mice compared to SPF mice (Fig 2A). Concentrations of 2-hydroxymetronidazole, primary metabolite, were only slightly decreased in GF mice compared to control SPF mice. Based on these results, plasma concentration of metronidazole could be affected by the gut microbiota, which lead to higher levels of the parent drug, metronidazole, in murine plasma of GF animals (in comparison to SPF ones). Studies have shown that metronidazole is weakly converted to the reduced metabolites (N-(2-hydroxyethyl)-oxamic acid and acetamide) by rat microbiota and these metabolites have been also found in human urine [27]. This could explain why there were higher plasma concentration of metronidazole observed in GF mice than in SPF mice without significant differences in concentrations of 2-hydroxymetronidazole.

The mRNA expression of Cyp2a5/4 was significantly increased after 6, respectively 24 hours in SPF mice, potentially leading to a more efficient metabolism of metronidazole in the liver, in GF mice it was decreased (Fig 4). However, the decrease of the metronidazole plasma level in the SPF mice (Fig 2A) cannot be simply explained by an effect of CYP2A enzymes, as

an increase of 2-OH metronidazole, main metabolite of metronidazole formed by CYP2A6 in human, is not prominent (Fig 2B). The most easy explanation of this may be that also other enzymes present in the SPF animals contributed to biotransformation of MTZ leading to lower plasma level of MTZ (Fig 2A). In this respect, a glucuronidation and possibly sulfation, both mentioned in Ref. 10, may contribute to the effect observed in SPF mice.

The results have shown that, beside direct bacterial metabolism, different expression and enzyme activity of hepatic biotransformation enzymes in the presence/absence of gut microbiota may be responsible for the altered metronidazole metabolism. Indeed, the differences in the expression and enzyme activity of some hepatic CYPs between GF and SPF mice have been shown earlier [28–30]. Interestingly, it was found that colonization by only a single non-pathogenic or probiotic bacteria strain alters the mRNA expression of some hepatic CYPs in originally GF mice [30]. Results of other studies have highlighted the role of gut microbiome and its metabolites in the regulation of CYPs expression, along with other factors [31, 32]. Drug administration may also contribute into this complex equation and influence the expression of CYPs, differently in the GF and SPF mice. This phenomenon has been seen in the case of anti-inflammatory drug nabumetone, where the differences in expression of CYPs in the small intestine and liver of GF and SPF mice after the drug administration were observed [33].

To contribute in a part to attempts aiming at uncovering the possible molecular changes leading to altered drug metabolism (here, it was metronidazole) in GF mice, the time course of the expression of transcription factors and CYPs was determined after metronidazole administration in comparison to the control mice. CYP genes are highly polymorphic and their expression is controlled by numerous genetic and nongenetic factors. Moreover, transcription factors such as AhR, CAR, PXR and PPARα are involved in their expression. Interestingly, the expression of transcription factors involved in the regulation of various CYPs has shown a different time course in samples from SPF and GF mice after metronidazole administration (Fig 3).

Interestingly, 24 hours after metronidazole application to the SPF mice, there was an evident increase of the CAR expression; yet in GF mice, metronidazole did not cause any significant changes of CAR mRNA level. Moreover, significant interaction between presence/absence of the gut microbiome and effect of metronidazole application was observed (Fig 3B). In SPF mice, the gut microbiota may have important role in influencing the size of bile acid pool and the absence of the microbiota (in GF mice) may result into increase of bile acid synthesis [34], which may lead in activation of CAR, as the bile acids act as a CAR ligand in the liver [35]. The fact that metronidazole decreased CAR mRNA levels more noticeably in GF mice (Fig 3B) could be caused by initially increased amounts of bile acids, as metronidazole was able to reduce biliary secretion of bile acids and cholesterol [36]. Interestingly, in SPF mice significant increase of CAR in 24th hour after application of metronidazole was observed. This could be result of altered microbiome composition (during dysbiosis) after metronidazole as it could affect bile acid pool, causing increased expression of CAR. The fact that mRNA expression of PXR and Nrf2 was significantly affected by administration of metronidazole and by presence (absence) of microbiome (Fig 3C, 3E) may contribute to explanation of the difference in the CYP enzymes mRNA expression in GF and SPF mice treated by metronidazole.

As it was mentioned earlier, the focus in this work has been on the CYP enzymes belonging to 1, 2 and 3 families, which are involved in the metabolism of drugs and xenobiotics. Various CYPs were generally more influenced by metronidazole in SPF mice than in the GF mice. Furthermore, metronidazole is known to alter the gut microbiota as it has been reported to slightly decrease the Firmicutes to Bacteroidetes ratio (due to the increase of Bacteroidetes) [37].

Significant combined effect of microbiome and metronidazole application was found also in Cyp2b10 mRNA expression (Fig 4C). In the absence of the gut microbiota (GF mice), mRNA expression of Cyp2b10 was significantly affected by metronidazole application.

Interestingly, mRNA level at 24th hour dropped to the control level. Regardless of significant increase of mRNA and increase of protein expression (Table 2), the activity of CYP2B was, however, not affected in GF mice (Fig 6C). In SPF mice, on the other hand, there was an evident trend of increasing enzyme activity of CYP2B (Fig 6C) after 24 hours of metronidazole administration, which correlated with the increase of mRNA (Fig 4C) and protein expression (Table 2) of CYP2B10. It is known, that human CYP2B6 (orthologue of murine CYP2B10) is inducible by several drugs (rifampicin, phenobarbital, carbamazepine and more) and it is possible that metronidazole can also affect the mRNA expression, protein expression and enzyme activity of murine CYP2B10 in the presence of gut microbiota. In addition, in both groups of animals (SPF and GF), upregulation of Cyp2b10 could be result of increased expression of PXR (Fig 3C) in 2nd hour after metronidazole application. Studies revealed, that CYP2B induction is likely co-regulated by both PXR and CAR [38]. As mentioned earlier, there is a possible effect of metronidazole on the gut microbiota, as the mRNA expression was not lowered 24 hours after administration of metronidazole in SPF mice, contrary to what has been seen in GF ones.

In SPF mice, another CYP enzyme (its mRNA expression and enzyme activity) affected by metronidazole administration was CYP2A. After the first use of metronidazole in 1959, it took decades to identify CYP2A6 as the enzyme responsible for metronidazole 2-hydroxylation in human [12]. It is still unknown if or how metronidazole is able to affect its own biotransformation. Stancil and colleagues [39] treated primary human hepatocytes with metronidazole, which resulted into an increase of CYP2A6 mRNA level and activity. In accordance with that result, it was found here that metronidazole along with gut microbiota (SPF mice) increased the mRNA expression of murine Cyp2a5/4 (Fig 4B). Interestingly, the activity of CYP2A was decreased by 50% in 2nd and 24th hour after drug administration, which demonstrate complexity of *in vivo* experiments. In GF mice, the CYP2A activity was also affected as we observed decrease in 2nd and 6th after metronidazole administration. Moreover, GF mice have shown lower level of enzyme activity of CYP2A than their SPF counterparts (Fig 6B), which can partially explain the more efficient metabolism of metronidazole in the liver of SPF mice. Based on these results, there was a possibility of potential time-dependent inhibition (formation of inhibitory metabolites or mechanism-based inhibition) of CYP2A. Pooled murine microsomes were pre-incubated with metronidazole (100 μmol/L) at two different times (2 and 30 min), however, there was no difference in CYP2A activity. According to the results, the effect of metronidazole on the CYP2A activity needs to be further study.

Our findings that Cyp3a11 mRNA expression and enzyme activity of CYP3A were lower in GF than SPF mice are in line with our previous studies [30, 31], showing that the presence of microbiome is crucial in the synthesis of CYP3A. However, it is not clear if this phenomenon contributes to more efficient metabolism of metronidazole in SPF mice.

The results presented here highlighted the effect of gut microbiota along with concomitantly taken medication on the pathways involved in the regulation of CYP synthesis and hepatic drug metabolism with apparently relevant clinical implications. Our data illustrate a complex interplay of both the altered properties of drug metabolism and the role of (also altered) gut microbiota. In light of this, further studies with focus on individual drugs and their combination are needed to better understand the ways gut microbiota alters drug metabolism and to increase the efficacy of the pharmacotherapy.

## Supporting information

**S1 Data.**
(DOCX)

## Author Contributions

**Conceptualization:** Eva Anzenbacherová.

**Data curation:** Tomáš Hudcovic.

**Formal analysis:** Nina Zemanová, Kateřina Lněničková, Markéta Vavrečková, Iveta Zapletalová, Petra Hermanová, Tomáš Hudcovic, Lenka Jourová.

**Methodology:** Petra Hermanová, Tomáš Hudcovic.

**Project administration:** Eva Anzenbacherová, Hana Kozáková.

**Supervision:** Eva Anzenbacherová.

**Validation:** Nina Zemanová, Kateřina Lněničková.

**Writing – original draft:** Nina Zemanová.

**Writing – review & editing:** Kateřina Lněničková, Eva Anzenbacherová, Pavel Anzenbacher, Hana Kozáková, Lenka Jourová.

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
