## [Decision Letter · Decision Letter 0]

29 Jun 2021

PONE-D-21-19140

Gut microbiome affects the metabolism of metronidazole in mice through regulation of hepatic cytochromes P450 expression

PLOS ONE

Dear Dr. Jourová,

Thank you for submitting your manuscript to PLOS ONE. After careful consideration, we feel that it has merit but does not fully meet PLOS ONE’s publication criteria as it currently stands. Therefore, we invite you to submit a revised version of the manuscript that addresses the points raised during the review process.

The reviewers pointed to information missing in the Materials and Methods section (e.g., Western blots should be presented following the PLOSone reporting requirements, etc.), potentially inappropriate statistical analyses, and an overinterpretation of the data (e.g., lack of consideration of other drug metabolizing enzymes).  I also concur with the suggestions from both reviewers regarding the presentation of the data. 

We look forward to receiving your revised manuscript.

Kind regards,

Hans-Joachim Lehmler, PhD

Academic Editor

PLOS ONE

4. We note that Figure 1 in your submission contain copyrighted images. All PLOS content is published under the Creative Commons Attribution License (CC BY 4.0), which means that the manuscript, images, and Supporting Information files will be freely available online, and any third party is permitted to access, download, copy, distribute, and use these materials in any way, even commercially, with proper attribution. For more information, see our copyright guidelines: http://journals.plos.org/plosone/s/licenses-and-copyright.

Reviewers' comments:

Reviewer's Responses to Questions

**Comments to the Author**

1. Is the manuscript technically sound, and do the data support the conclusions?

Reviewer #1: Partly

Reviewer #2: Yes

2. Has the statistical analysis been performed appropriately and rigorously? 

Reviewer #1: No

Reviewer #2: No

3. Have the authors made all data underlying the findings in their manuscript fully available?

Reviewer #1: Yes

Reviewer #2: No

4. Is the manuscript presented in an intelligible fashion and written in standard English?

Reviewer #1: Yes

Reviewer #2: No

5. Review Comments to the Author

Reviewer #1: This manuscript has used germ free (GF) and specific pathogen free (SPF) to demonstrate that the absence of gut microbiota appeared to result in a temporal trend of increase (although not statistically significant) in the plasma metronidazole 2 hours after oral administration, but not at other time points. The hydroxylated metabolite determined was not changed. In addition, this study demonstrated that oral metronidazole the mRNA expression of selected P450s (CYP1-3) and xenobiotic-sensing transcription factors to a different extent between SPF and GF mice, although common patterns were also observed such as an up-regulation of Cyp2b10 by metronidazole in both mouse groups, over the time course. In contrast, the protein and enzyme activities were not readily altered significantly, although certain trend of increase or decrease was observed.

Major points:

1. Figure 2 – Figure 5: it is strongly recommended that the SPF and GF data should be plotted in the same figure, so one can evaluate the basal differences between SPF and GF. If this results in too many bars in one figure, the plots can be split per time point. This is important because it is highly possible that the tendency in the change in the metronidazole pharmacokinetics is not due to the differential response patterns of the P450s to metronidazole treatment between SPF and GF, but simply due to the basal differences in these enzymes between SPF and GF.

2. Please show the western blot result as the actual immuno-blots (currently Table 1), so that one can evaluate the specificity of the bands, sample size, and biological variations. Please include the standard deviations or standard errors in Table 1.

3. Method section: The RT-qPCR normalization method claimed that the data were expressed as ddCq and normalized to the Hprt house-keeping gene. However, It looks like Figure 3 and Figure 4 actually artificially assigned both SPF control and GF control to 1. This might not be the best way to express the data because the SPF control may be different from the GF control. Please consider re-plotting the data so it matches the Method section.

4. Method section: statistical method: one-way ANOVA was performed. However, this study design has two factors, namely enterotype (SPF vs. GF) and treatment (control vs. metronidazole over a time course). Thus two-way ANOVA is a more appropriate method.

5. From the plasma metronidazole result (Figure 2A), I think the 2h time point is the most critical for evaluation regarding mRNA expression, protein levels, and enzyme activities. Please especially take a look into this time point to connect the Figure 2A result with the other figures.

6. Figure 2A and 2B: because the hydroxylated metabolite did not decrease to a similar extent as the apparent increase of the parent compound, it is possible that this may be a disposition issue. Please consider measuring hepatic metronidazole and its metabolite (and maybe even urine). Also one needs to consider transporters and Ugts in addition to the P450s.

7. Introduction section says CYP2A6 in humans is the most predominant enzyme for metronidazole metabolism. However the mouse data on the Cyp2a isoforms do not seem to explain the trend of increase in plasma metronidazole at 2 hours; in fact there was a clear trend of decrease in CYP2A enzyme activities to a greater extent in SPF mice than in GF mice (opposite to the expectations) at 2h. The mRNA increase in Cyp2a was only observed after 6h and is not likely responsible for the increase in plasma metronidazole at 2h. Would other P450 isoforms be responsible for the mouse metabolism of this drug? Please discuss based on your findings.

Minor points:

1. Method section: Western blot: please include the catalog numbers of the antibodies.

2. Method section: please define the age, sex, and sample size of the mice.

3. Abstract: please be more specific in describing the key findings on mRNA, protein, and enzyme activities.

4. Title: it seems that the study was mainly determining the effect of the absence of gut microbiome on the hepatic P450s, not necessarily the effect of the gut microbiome on the hepatic P450s.

5. Typo: “could not be determinate.” → “ could not be determined.”

Reviewer #2: Manuscript ID: PONE-D-21-19140

Title: Gut microbiome affects the metabolism of metronidazole in mice through regulation of hepatic cytochromes P450 expression

The manuscript described the effect of gut microbiome on the fate of metronidazole by profiling the metronidazole’s pharmacokinetics changes (Cmax and metabolic ratio). And the selected P450 enzymes related changes (mRNA expression, protein expression and enzyme activity) were compared in both SPF and germ-free (GF) mice after administration metronidazole within 24 hours. And the results provided the possibility of the selected hepatic P450 enzymes altered metronidazole metabolism.

1. There were some wording issues e.g. CYP (line171) and one cited literature [36] information about metronidazole has not been found within the mentioned literature content.

2. The structure and enzymatic information that mentioned in the introduction section was not clearly marked in the Figure 1.

3. Some antibiotics mechanism that mentioned in the manuscript (Introduction section) has been summarized in the related review [1] and should be mentioned in the introduction.

4. A lot of details of Section ‘cytochrome P450 enzyme activity assays’ were missing. Such as the particle size of column; the UV/ fluorescence detection parameters, and the gradient ranges.

5. The metabolic ratios were mentioned with only the values (line 234-238). Did the metabolic ratio calculate individually? What is the mean and SD for the metabolic ratio?

6. Could the Figure 3-5 changed into a box plot with individual values plus mean and SD to facilitate the data trend observation?

7. Literature has reported that co-dosing metronidazole with neomycin, vancomycin, and ampicillin for 3 weeks could obtain pseudo germ-free mice [2]. Could the short-term treatment with metronidazole in SPF mice regulate similar metabolic pathway? If so, did the comparison between SPF mice and GF mice after administration metronidazole describe the difference between pseudo germ-free mice versus germ-free mice metabolic pathway?

8. The pharmacokinetic data showed in Figure 2 have a clear overlap in Cmax value. Especially, the SD of the GF mice Cmax approached the SPF mice Cmax mean value. Was there a possibility that there is an outlier plasma concentration within the limited sample numbers? Could the individual Cmax values be attached as supplementary data?

9. Besides the described results, were there any RNA-seq screening data and/or proteomics data results support the selected metabolic enzyme regulation? Or does there any other metabolic enzyme also significantly involved into the metronidazole metabolism?

Reference

[1] Leitsch D. A review on metronidazole: an old warhorse in antimicrobial chemotherapy. Parasitology. 2019 Aug;146(9):1167-78.

[2] Liang W, Zhao L, Zhang J, Fang X, Zhong Q, Liao Z, Wang J, Guo Y, Liang H, Wang L. Colonization Potential to Reconstitute a Microbe Community in Pseudo Germ-Free Mice After Fecal Microbe Transplant From Equol Producer. Frontiers in Microbiology. 2020 Jun 5;11:1221.

6. PLOS authors have the option to publish the peer review history of their article (what does this mean?). If published, this will include your full peer review and any attached files.

Reviewer #1: No

Reviewer #2: No

---

## [Author Response · Author response to Decision Letter 0]

23 Aug 2021

Major points: 

1. Figure 2 – Figure 5: it is strongly recommended that the SPF and GF data should be plotted in the same figure, so one can evaluate the basal differences between SPF and GF. If this results in too many bars in one figure, the plots can be split per time point. This is important because it is highly possible that the tendency in the change in the metronidazole pharmacokinetics is not due to the differential response patterns of the P450s to metronidazole treatment between SPF and GF, but simply due to the basal differences in these enzymes between SPF and GF. 

Response: Thanks for comments, the reviewer is right. We did a new statistical analysis using two-way ANOVA to show more clearly the effect of two variables (presence of the microbiome, and effect of metronidazole in time). This method allowed us to show the difference of the basal expression between SPF and GF mice. As it can be seen from the results, there is indeed the difference between basal expression in GF and SPF mice in some genes and in some there is not. However, the significant effect of metronidazole administration itself has been proved in the most of the selected genes.

2. Please show the western blot result as the actual immuno-blots (currently Table 1), so that one can evaluate the specificity of the bands, sample size, and biological variations. Please include the standard deviations or standard errors in Table 1. 

Response: In fact, the Western blots did not show prominent changes in protein content. The samples were pooled (N=4), hence the SD or statistical analysis (showing the biological variation) cannot be given. We assume that the results of protein expression do not explain simply the discrepancy between mRNA expression and enzyme activity (in the case of some CYPs), nor do they significantly contribute to the explanation of the effect of absence/presence of gut microbiome on the plasma concentration of the metronidazole. These were also the reasons why we decided not to show them in detail. We enclose the results of representative experiments as a Figure 5 (showing mainly an increase of the CYP2B10 in SPF mice). 

3. Method section: The RT-qPCR normalization method claimed that the data were expressed as ddCq and normalized to the Hprt house-keeping gene. However, It looks like Figure 3 and Figure 4 actually artificially assigned both SPF control and GF control to 1. This might not be the best way to express the data because the SPF control may be different from the GF control. Please consider re-plotting the data so it matches the Method section. 

Response: In the line with the response to question 1 we re-plotted the data using two-way ANOVA statistical analysis to show also the difference between basal expression in GF and SPF mice.

4. Method section: statistical method: one-way ANOVA was performed. However, this study design has two factors, namely enterotype (SPF vs. GF) and treatment (control vs. metronidazole over a time course). Thus two-way ANOVA is a more appropriate method. 

Response: We thank the reviewer for suggestion concerning statistical analysis. We analysed the data using two-way ANOVA which helped significantly to explore the combined effect of gut microbiome and metronidazole administration on the mRNA expression of selected transcriptional factors and cytochromes P450 and to correctly present obtained data. We added this fact also in the Materials and Method section. We thank the reviewer again for the comments which helped to present the data correctly and significantly improved the manuscript. 

5. From the plasma metronidazole result (Figure 2A), I think the 2h time point is the most critical for evaluation regarding mRNA expression, protein levels, and enzyme activities. Please especially take a look into this time point to connect the Figure 2A result with the other figures. 

Response: After careful inspection of the data, we feel that a direct relation between an increase of the metronidazole plasma level after 2 hours and expression of CYP enzymes is difficult. In the SPF mice, also the effects of other processes contributing to lowering of the metronidazole level in presence of microbiome cannot be ruled out. 

6. Figure 2A and 2B: because the hydroxylated metabolite did not decrease to a similar extent as the apparent increase of the parent compound, it is possible that this may be a disposition issue. Please consider measuring hepatic metronidazole and its metabolite (and maybe even urine). Also one needs to consider transporters and Ugts in addition to the P450s. 

Response: We wish to thank the reviewer for this comment. Reviewer is correct, as there could be possibility that it can be a disposition issue. Unfortunately, we are not able to measure the hepatic metronidazole and its metabolite due to scarcity of material, as we primary focused on the activity assays of CYP enzymes (obtained from hepatic microsomes). Other transporters or enzymes of II. phase of biotranformation may play a role in metronidazole metabolism/disposition. As mentioned in Discussion, glucuronidation and possibly sulfation can be involved, however, to a much lesser extent than hydroxylation and oxidation of metronidazole. For this reason, and due to scarcity of material, we focused only on cytochromes P450 determination.

7. Introduction section says CYP2A6 in humans is the most predominant enzyme for metronidazole metabolism. However the mouse data on the Cyp2a isoforms do not seem to explain the trend of increase in plasma metronidazole at 2 hours; in fact there was a clear trend of decrease in CYP2A enzyme activities to a greater extent in SPF mice than in GF mice (opposite to the expectations) at 2h. The mRNA increase in Cyp2a was only observed after 6h and is not likely responsible for the increase in plasma metronidazole at 2h. Would other P450 isoforms be responsible for the mouse metabolism of this drug? Please discuss based on your findings. 

Response: The reviewer is right. We thank for the comment and apologize for inappropriate interpretation/explanation of the obtained data. Indeed, it is not possible to explain the change in plasma level of metronidazole simply by changes in the Cyp2a5/4 expression. While it has been reported that in humans, CYP2A6 is responsible for hydroxylation of metronidazole, its metabolism in mice is still unknown and possibly another enzymes may contribute to the biotransformation.

Moreover, the time course of the processes as expression of mRNA, expression of proteins and changes of activities are not easy to explain. As the effect of the presence of microbiome in SPF mice on the level of the metabolite is relatively less pronounced (in comparison with the decrease of the metronidazole level which is more prominent, cf. Fig. 2A and 2B), it may be, that there are other processes contributing to lowering of the metronidazole levels. Namely, an effect of glucuronidation and sulfation should be taken in account (both mentioned in ref. 10). This possibility has been also implemented to the Discussion (lines 367 – 375).

Minor points:

1. Method section: Western blot: please include the catalog numbers of the antibodies. 

Response: The catalog numbers of the antibodies were added to the manuscript (lines 190 – 197).

2. Method section: please define the age, sex, and sample size of the mice. 

Response: Description (age, sex and sample size) of used animals is mentioned in the Materials and Method section (Animals) - lines 128 – 130.

3. Abstract: please be more specific in describing the key findings on mRNA, protein, and enzyme activities.

Response: We rewrote the abstract. We would thank the reviewer for this comment.

 4. Title: it seems that the study was mainly determining the effect of the absence of gut microbiome on the hepatic P450s, not necessarily the effect of the gut microbiome on the hepatic P450s. 

Response: We thank the reviewer for this suggestion, however, we would rather keep our first suggested title. It depends on which group we take as a control (SPF or GF). When we compare between the two groups we study the effect of presence/absence of the gut microbiota on selected metabolic pathways. For simplicity and conciseness of the title, we prefer to keep the original title.

5. Typo: “could not be determinate.” � “ could not be determined.”

Response: We apologize for this mistake. It has been rewritten. (line – 215).

 

1. There were some wording issues e.g. CYP (line171) and one cited literature [36] information about metronidazole has not been found within the mentioned literature content.

Response: This was a mistake we made in the preparation of the manuscript and we have corrected it (wording issue in line 171). 

The information about metronidazole was found in paper’s (Zhang et al., 2014) results; subchapter - Antibiotics treatment restructured bacterial community – “Metronidazole slightly decreased the Firmicutes to Bacteroidetes ratio, mainly due to the increase in Bacteroidetes.”

2. The structure and enzymatic information that mentioned in the introduction section was not clearly marked in the Figure 1. 

Response: Thanks to the reviewer for the comment. Cytochrome P450 2A6 (known to be responsible for hydroxylation of metronidazole in human) was implemented in the Figure 1. And the information about bacterial reduction of metronidazole was added to the Introduction (line 96– 99).

3. Some antibiotics mechanism that mentioned in the manuscript (Introduction section) has been summarized in the related review [1] and should be mentioned in the introduction.

Response: We have added this reference to our paper (reference 12 in Introduction – line 90).

4. A lot of details of Section ‘cytochrome P450 enzyme activity assays’ were missing. Such as the particle size of column; the UV/ fluorescence detection parameters, and the gradient ranges.

Response: The authors are grateful for this remark. We have added additional information to the columns used for the activity assays and we have also added the Table 1 with information about elution and detection (lines 200 – 205).

5. The metabolic ratios were mentioned with only the values (line 234-238). Did the metabolic ratio calculate individually? What is the mean and SD for the metabolic ratio?

Response: We wish to thank the reviewer for this comment. The metabolic ratio was not calculated individually for each animal, but rather it was expressed as ratio of the means (of the unchanged drug – metronidazole, to its metabolite – 2-hydroxymetronidazole) at 2nd and 6th hour. This is the reason why the values are expressed without SD.

6. Could the Figure 3-5 changed into a box plot with individual values plus mean and SD to facilitate the data trend observation?

Response: Thank you for this suggestion. However, the figures 3 and 4 (mRNA expression of transcription factors and CYPs) have been corrected using two-way ANOVA as a more appropriate method which led us to another type of graphs. 

7. Literature has reported that co-dosing metronidazole with neomycin, vancomycin, and ampicillin for 3 weeks could obtain pseudo germ-free mice [2]. Could the short-term treatment with metronidazole in SPF mice regulate similar metabolic pathway? If so, did the comparison between SPF mice and GF mice after administration metronidazole describe the difference between pseudo germ-free mice versus germ-free mice metabolic pathway? 

Response: This is a very interesting remark. The objective of the mentioned study was to establish human microbiota-associated mice model for equol production through pseudo germ-free mice, mimicking the gut microbiota of an adult human equol producer. However, we are not sure if there is a possibility that short-term treatment with metronidazole in SPF mice could regulate similar metabolic pathways, as they used combination of vancomycin, neomycin sulfate, metronidazole and ampicillin to obtain pseudo germ-free mice. 

8. The pharmacokinetic data showed in Figure 2 have a clear overlap in Cmax value. Especially, the SD of the GF mice Cmax approached the SPF mice Cmax mean value. Was there a possibility that there is an outlier plasma concentration within the limited sample numbers? Could the individual Cmax values be attached as supplementary data?

Response: The reviewer is right there is an overlap in Cmax values. We suggest, that the SD values are affected by the differences among the animals, as this phenomenon was noticeable in every in vivo experiment we performed in our laboratory and we are not sure, if increased number of animals would solve this problem. We have provided the individual Cmax values in the Supplemental Data file. (mentioned in the manuscript - lines 239 – 240).

9. Besides the described results, were there any RNA-seq screening data and/or proteomics data results support the selected metabolic enzyme regulation? Or does there any other metabolic enzyme also significantly involved into the metronidazole metabolism?

Response: This is a good suggestion, however besides the described results, there are no results such as RNA-sew screening or proteomics data. 

Glucuronidation and possibly sulfation is known to be involved, however, to a much lesser extent than hydroxylation and oxidation of metronidazole. Besides cytochrome P450 2A6 (in human), no specific enzyme has been reported to metabolize metronidazole.

---

## [Decision Letter · Decision Letter 1]

11 Oct 2021

PONE-D-21-19140R1Gut microbiome affects the metabolism of metronidazole in mice through regulation of hepatic cytochromes P450 expressionPLOS ONE

Dear Dr. Jourová,

Thank you for submitting your manuscript to PLOS ONE. After careful consideration, we feel that it has merit but does not fully meet PLOS ONE’s publication criteria as it currently stands. Therefore, we invite you to submit a revised version of the manuscript that addresses the points raised during the review process.  Some major concerns must be addressed.

Please submit your revised manuscript within 60 days. If you will need more time than this to complete your revisions, please reply to this message or contact the journal office at plosone@plos.org. Please include the following items when submitting your revised manuscript:A rebuttal letter that responds to each point raised by the academic editor and reviewer(s). You should upload this letter as a separate file labeled 'Response to Reviewers'.A marked-up copy of your manuscript that highlights changes made to the original version. You should upload this as a separate file labeled 'Revised Manuscript with Track Changes'.An unmarked version of your revised paper without tracked changes. You should upload this as a separate file labeled 'Manuscript'.

We look forward to receiving your revised manuscript.

Kind regards,

Gianfranco D. Alpini

Academic Editor

PLOS ONE

Journal Requirements:

Reviewers' comments:

Reviewer's Responses to Questions

**Comments to the Author**

1. If the authors have adequately addressed your comments raised in a previous round of review and you feel that this manuscript is now acceptable for publication, you may indicate that here to bypass the “Comments to the Author” section, enter your conflict of interest statement in the “Confidential to Editor” section, and submit your "Accept" recommendation.

Reviewer #2: All comments have been addressed

Reviewer #3: (No Response)

2. Is the manuscript technically sound, and do the data support the conclusions?

Reviewer #2: (No Response)

Reviewer #3: Yes

3. Has the statistical analysis been performed appropriately and rigorously? 

Reviewer #2: (No Response)

Reviewer #3: Yes

4. Have the authors made all data underlying the findings in their manuscript fully available?

Reviewer #2: (No Response)

Reviewer #3: Yes

5. Is the manuscript presented in an intelligible fashion and written in standard English?

Reviewer #2: (No Response)

Reviewer #3: Yes

6. Review Comments to the Author

Reviewer #2: (No Response)

Reviewer #3: In the revised version of the manuscript, the authors have responded some comments and added some new data. However, there are still several concerns should be addressed before it can be considered for publication.

1. Better and individual animal western blot bands are requested.

2. Figures should be arranged in order with results.

3. Some results and Figures (order) are not match.

4. Missing X-axis in Figure 4 C and G.

7. PLOS authors have the option to publish the peer review history of their article (what does this mean?). If published, this will include your full peer review and any attached files.

Reviewer #2: No

Reviewer #3: No

---

## [Author Response · Author response to Decision Letter 1]

13 Oct 2021

Dear Editor, 

thank You for your kind revision and suggestions for improvement of our paper. We wish to thank reviewers as well for their helpful comments and for taking the time to point out options to improve our manuscript. 

Below, we have addressed the comments of Reviewer #3: 

1. Better and individual animal western blot bands are requested.

Unfortunately, we are not able to show Western blots for individual animals. The reason is that the experiment has been realized with SPF and germ-free (GF) mice to document the differences in response caused of presence of the microbiome. In fact, the GF animal model is very unique (and only few laboratories in the world have access to this model) and we were limited by number of animals and by the amount of experimental material (small murine livers). On the other hand, there was the need to get the information on the activities of individual CYP forms. This is why we pooled the samples to get as much information as possible (enzyme activities, protein and mRNA expression). We deal with this issue in every experiment with murine livers obtained with the germ-free models. We feel that even when the microsomes are pooled, the results and corresponding figures support the conclusions reached.

2. Figures should be arranged in order with results.

We thank very much the Reviewer for bringing this issue to our attention. We moved the legends of the figures after the corresponding description in the results.

3. Some results and Figures (order) are not match.

Thank you for the comment. We apologize for the mistake and we corrected the number of the figure in the manuscript. 

4. Missing X-axis in Figure 4 C and G.

The reviewer is right that the figure C and G differ from the other graphs. However, description of X-axis is not missing here as it is only in a different position due to SD reaching the minus values. 

We would like to say again that we appreciate all your time given to our article. 

With best regards

Lenka Jourová, Ph.D.

Department of Medical Chemistry and Biochemistry

Faculty of Medicine and Dentistry, Palacký University

Hněvotínská 3

775 15 Olomouc, Czech Republic

Tel: +420 585 632 346, email: LenkaJourova@seznam.cz

---

## [Decision Letter · Decision Letter 2]

25 Oct 2021

Gut microbiome affects the metabolism of metronidazole in mice through regulation of hepatic cytochromes P450 expression

PONE-D-21-19140R2

Dear Dr. Jourová,

We’re pleased to inform you that your manuscript has been judged scientifically suitable for publication and will be formally accepted for publication once it meets all outstanding technical requirements.

Kind regards,

Gianfranco D. Alpini

Academic Editor

PLOS ONE

Additional Editor Comments (optional):

Reviewers' comments:

Reviewer's Responses to Questions

**Comments to the Author**

1. If the authors have adequately addressed your comments raised in a previous round of review and you feel that this manuscript is now acceptable for publication, you may indicate that here to bypass the “Comments to the Author” section, enter your conflict of interest statement in the “Confidential to Editor” section, and submit your "Accept" recommendation.

Reviewer #3: All comments have been addressed

2. Is the manuscript technically sound, and do the data support the conclusions?

Reviewer #3: Yes

3. Has the statistical analysis been performed appropriately and rigorously? 

Reviewer #3: Yes

4. Have the authors made all data underlying the findings in their manuscript fully available?

Reviewer #3: Yes

5. Is the manuscript presented in an intelligible fashion and written in standard English?

Reviewer #3: Yes

6. Review Comments to the Author

Reviewer #3: The authors addressed all of my previous concerns and I have no more comments or issues regarding this manuscript.

7. PLOS authors have the option to publish the peer review history of their article (what does this mean?). If published, this will include your full peer review and any attached files.

Reviewer #3: No

---

## [Editor Report · Acceptance letter]

29 Oct 2021

PONE-D-21-19140R2 

Gut microbiome affects the metabolism of metronidazole in mice through regulation of hepatic cytochromes P450 expression 

Dear Dr. Jourová:

I'm pleased to inform you that your manuscript has been deemed suitable for publication in PLOS ONE. Congratulations! Your manuscript is now with our production department. 

Kind regards, 

on behalf of

Dr. Gianfranco D. Alpini 

Academic Editor

PLOS ONE